# Adipokine Levels in Men with Coronary Atherosclerosis on the Background of Abdominal Obesity

**DOI:** 10.3390/jpm12081248

**Published:** 2022-07-29

**Authors:** Evgeniia Vital’evna Striukova, Victoriya Sergeevna Shramko, Elena Vladimirovna Kashtanova, Yana Vladimirovna Polonskaya, Ekaterina Mikhailovna Stakhneva, Alexey Vitalievich Kurguzov, Alexander Mikhailovich Chernyavsky, Yulia Igorevna Ragino

**Affiliations:** 1Research Institute of Internal and Preventive Medicine–Branch of the Institute of Cytology and Genetics, Siberian Branch of Russian Academy of Sciences (IIPM—Branch of IC&G SB RAS), B. Bogatkova Str., 175/1, 630089 Novosibirsk, Russia; nosova@211.ru (V.S.S.); elekastanova@yandex.ru (E.V.K.); yana-polonskaya@yandex.ru (Y.V.P.); stahneva@yandex.ru (E.M.S.); ragino@mail.ru (Y.I.R.); 2The Federal State Budgetary Institution “National Medical Research Center Named Academician E.N. Meshalkin” of the Ministry of Health of the Russian Federation, Rechkunovskaya Str., 15, 630055 Novosibirsk, Russia; aleksey_kurguzov@mail.ru (A.V.K.); amchern@mail.ru (A.M.C.)

**Keywords:** obesity and coronary atherosclerosis, unstable atherosclerotic plaque, TNFa, IL-6, C-peptide

## Abstract

**Background.** Obesity is associated with dyslipidemia, and excess body fat is associated with unfavorable levels of adipokines and markers of inflammation. **The goal of research.** To study the level of adipokines and markers of inflammation, their associations with unstable atherosclerotic plaques in men with coronary atherosclerosis on the background of abdominal obesity. **Materials and methods.** The study involved 82 men aged 40–77 years with coronary atherosclerosis after endarterectomy from the coronary arteries. We divided all men into two groups: 37 men (45.1%) with unstable atherosclerotic plaques, and 45 men (54.9%) who had stable plaques. Obesity was established at a BMI of ≥30 kg/m^2^. The levels of adipokines and markers of inflammation in the blood were determined by multiplex analysis. **Results.** In patients with obesity and unstable plaques, the levels of C-peptide, TNFa and IL-6 were 1.8, 1.6, and 2.8 times higher, respectively, than in patients with obesity and stable plaques. The chance of having an unstable plaque increases with an increase in TNFa by 49% in obese patients and decreases with an increase in insulin by 3% in non-obese patients. **Conclusions.** In men with coronary atherosclerosis and obesity, unstable atherosclerotic plaques in the coronary arteries are directly associated with the level of TNF-α.

## 1. Introduction

Two billion people worldwide over the age of 18, or approximately 30% of the world’s population, are overweight or obese [1]. In Russia, from 2012 to 2018, the number overweight individuals increased by 7.8%, amounting to 40.3% [2]. In the Novosibirsk region, the prevalence of obesity was 35.0%; among men, it was 20.7%, and among women it was 47.0% [3].

Despite considerable efforts aimed at understanding the biology of obesity, it became obvious that the available knowledge has not helped to curb the obesity epidemic today and that no part of the world has been spared from this phenomenon, and therefore, the study of this problem is an urgent problem of modern endocrinology. The Global Burden of Disease group also estimated that elevated body mass index (BMI) values caused 4 million deaths in 2015, with two-thirds of this number due to cardiovascular diseases (CVD) [1].

Obesity is closely related to dyslipidemia, which is mainly due to insulin resistance and the effects of pro-inflammatory adipokines [4]. A large number of studies have shown that excess adipose tissue in the body is associated with unfavorable levels of adipokines and inflammatory markers; for example, interleukin-6 (IL-6) and tumor necrosis factor alpha (TNF-α), which can also participate in the formation of unstable atherosclerotic plaque [5]. C-peptide has chemotactic effect, which correlates with its participation in the early stages of atherogenesis, and glucagon and insulin may have a protective effect in the development of atherosclerotic plaque [6,7,8]. Therefore, the study of adipokines (hormones produced by adipocytes) and their effects is an essential direction in modern medicine.

Thus, the aim of our study was to study the levels of adipokines and markers of inflammation (C-peptide, glucose-dependent insulinotropic polypeptide (GIP), glucagon-like peptide-1 (GLP-1), glucagon, interleukin-6 (IL-6), insulin, leptin, monocyte chemoattractant protein-1 (MCP-1), tumor necrosis factor alpha (TNF-α)) as well as their associations with unstable atherosclerotic plaques in men with coronary atherosclerosis against the background of abdominal obesity.

## 2. Materials and Methods

The design of the study is a single-stage observational study. The study was conducted as part of a joint scientific research of IIPM—Branch of IC&G SB RAS and FSBI “National Medical Research Center named after E. Meshalkin” of the Ministry of Health of the Russian Federation. The study was conducted in accordance with the Declaration of Helsinki, and approved by the Local Ethics Committees of both institutions (IIPM—Branch of IC&G SB RAS and NMRC named after ak. E.N. Meshalkin) (protocol No. 2, approval on 5 July 2011). The data and samples were collected after receiving written informed consent from all participants of the study.

The study involved 130 men aged 40–77 years (mean age 54.58 ± 10.33). They had verified atherosclerosis of the coronary arteries, without acute coronary syndrome (ACS), with stable angina pectoris II-III FC, hospitalized in the clinic of the FSBI National Medical Research Center named after E. Meshalkin of the Ministry of Health of the Russian Federation for coronary bypass surgery (CABG), in the period from 2011 to 2021.

Inclusion criteria at the pre-selection stage of patients: male gender, diagnosis of coronary heart disease (CHD), verified by coronary angiography data, the presence a myocardial infarction (MI) history or episodes of stable angina pectoris, documented by a description of the clinical picture of the disease, the results of ECG and biochemical blood tests.

Exclusion criteria at the pre-selection stage of patients included the presence of ACS less than 6 months prior to admission (unstable angina or MI), clinically significant severe chronic diseases in acute stage (chronic infectious and inflammatory diseases, renal failure, respiratory failure, liver failure), known active oncological diseases, and toxic damage by heavy metals.

During CABG surgery, 82 patients, with an average age of 51.91 ± 11.034, underwent endarterectomy from the coronary artery(s) strictly according to intraoperative indications. Further studies of histological material were carried out in the pathomorphological laboratory of the FSBI “NMRC im. Academician E.N. Meshalkin of the Ministry of Health of Russia”. Each endarterectomy material with intima-media of the coronary arteries was longitudinally and transversely symmetrically divided into 3–5 fragments for histological and biochemical analysis. Histological analysis of fragments after macroscopic description of samples (prevalence of atherosclerotic plaque, degree of narrowing of the artery lumen, hemorrhages in the structures of atherosclerotic plaque, calcification sites, blood clots) and standard hematoxylin-eosin and Van Gieson staining were studied on an Axiostar Plus binocular microscope. The study of fragments of intimate media revealed the presence of stable and unstable atherosclerotic plaques. The unstable plaque was differentiated according to the following criteria: the thickness of the fibrous covering was less than 65 microns, infiltration by macrophages and T-lymphocytes (over 25 cells in the field of vision 0.3 mm), a large lipid nucleus (over 40%) [9]. Of the studied group of patients, 45 men had stable plaques in the coronary arteries (54.9%), and 37 men had unstable plaques in the coronary arteries (45.1%) (Figure 1).

The study took into account demographic characteristics, past medical history, the presence of chronic diseases (type 2 diabetes, hypertension, myocardial infarction, acute cerebrovascular event). Patients underwent anthropometry, including measurements of height, body weight, waist, and hip circumference. The body mass index was determined by the following formula: BMI = Body weight (kg)/Height (m^2^). Waist circumference was measured in a standing position, in the middle of the distance between the lower edge of the chest and the crest of the ilium along the middle axillary line. Obesity was established at a BMI ≥ 30 kg/m^2^ [10]. Among all patients with obesity 23 (67.6%) also had criteria of abdominal obesity—waist circumference ≥ 94 cm.

BP was measured three times with an interval of two minutes on the right arm in a sitting position after a 5-min rest by using an automatic tonometer Omron M5-I with the registration of the average value of 3 measurements. Arterial hypertension (AH) was determined at systolic blood pressure (SBP) ≥ 140 mmHg and/or diastolic blood pressure (DBP) ≥ 90 mmHg.

Before the operation, biological material (blood) was taken. Blood tests were carried out in the Laboratory of Clinical Biochemical and Hormonal studies of Therapeutic Diseases of IIPM—Branch of IC&G SB RAS. By multiplex analysis using the Human Metabolic Hormone V3 panel (MILLIPLEX, Germany), levels of adipokines and inflammatory markers were determined on a Luminex MAGPIX flow fluorimeter: C-peptide, glucose-dependent insulinotropic polypeptide (GIP), glucagon-like peptide-1 (GLP-1), glucagon, interleukin-6 (IL-6), insulin, leptin, monocytic chemoattractant protein-1 (MCP-1), and tumor necrosis factor alpha (TNF-α).

Statistical processing was made by using the SPSS 13.0 software package. To assess the distribution patterns of signs, we used the Kolmogorov–Smirnov test. A comparative study of clinical and anamnestic characteristics in the groups was carried out by using the Student’s *t*-test. In the text, these characteristics of features are presented in the form of an arithmetic mean (M) and a standard deviation (SD). Under normal distribution, calculations were performed by using the ANOVA test. In the case of an abnormal distribution, the nonparametric Mann–Whitney U test (for two independent groups) were used. Spearman’s rank correlation coefficient (r s) was used to analyze the dependence of quantitative features of sample data from aggregates. In the case of nominal and ordinal data scales, cross-data tables and Pearson’s χ^2^ criterion were used, adjusted for probability (for calculating ORs). The significance level was also set at ***p*** < 0.05.

## 3. Results

Table 1 presents the characteristics of population and patients depending on the type of plaque (stable/unstable) and the presence of obesity. Groups of men with obesity and without it were comparable in age, SBP, DBP, smoking status, the presence of type 2 diabetes regardless of the type of plaque. All patients had an established diagnosis of hypertension, took antihypertensive drugs until the target blood pressure values were reached.

Based on the very high risk of all patients included in the study, all patients with coronary artery disease received high-intensity statin therapy in the maximum tolerated dosages (rosuvastatin 20–40 mg, atorvastatin 40–80 mg).

Table 2 shows the content of adipokines and inflammatory markers, depending on obesity in patients with stable and unstable atherosclerotic plaques.

When analyzing subgroups of patients depending on the type of plaque and obesity, no differences were obtained between patients with stable plaques with or without obesity. For a subgroup of patients with unstable plaques, the difference between obese and non-obese patients in the levels of C-peptide, GIP, GLP-1, interleukin-6, insulin, and leptin was obtained.

Figure 2 shows the content of C-peptide, IL-6, TNFa in the blood of obese patients, depending on the type of plaque.

Between patients with obesity and stable plaques and with obesity and unstable plaques, differences were obtained for C-peptide (*p* = 0.046), TNFa (*p* = 0.0042), and the trend for IL-6 (0.05).

Figure 3 shows the content of glucagon, insulin in the blood of patients without obesity, depending on the type of plaque.

Differences were obtained between patients without obesity and stable plaques and without obesity and unstable plaques for glucagon (*p* = 0.040), insulin (*p* = 0.011).

At the next stage, we included all the studied parameters in the logistic regression analysis model of the chance of having an unstable plaque depending on the studied parameters (Table 3).

The results showed that the chance of having an unstable plaque increases with an increase in TNFa by 49% in obese patients. The chance of having an unstable plaque decreases with an increase in insulin by 3% in non-obese patients.

## 4. Discussion

The development of atherosclerosis and the subsequent destabilization of atherosclerotic plaques are the main pathology underlying (ischemic) heart disease. Therefore, early detection of unstable atherosclerotic plaques by using biomarkers may be useful for reducing the frequency of acute cardiovascular syndromes.

C-peptide is a cleavage product of proinsulin, which acts on various types of cells. C-peptide has ambiguous effects, acting as an antithrombotic or thrombotic molecule, depending on the physiological environment and conditions of the disease [11] In addition, C-peptide regulates the interaction of leukocytes, erythrocytes and platelets with the endothelium [12]. Positive effects include stimulation of nitric oxide production with its subsequent release by platelets and endothelium [13], interaction with erythrocytes leading to the formation of adenosine triphosphate, and inhibition of the release of atherogenic cytokines [14]. The undesirable effect of C-peptide includes chemotaxis of monocytes, lymphocytes, and smooth muscle cells [6]. In several studies, the correlation of C-peptide with the proatherogenic process has been established. By using immunohistochemical analysis of postmortem arteries, Marx et al. showed the deposition of C-peptide in the intima and subendothelium of diabetic patients, as well as the co-deposition of C-peptide with 96% of all CD68+ monocytes and macrophages, which are responsible for the absorption of oxidized low-density lipoprotein, forming foam cells. These data indicate the chemotactic effect of C-peptide, which, therefore, correlates with its participation in the early stages of atherogenesis [6]. Also, C-peptide was associated with an increase in lipid deposits and increased proliferation of smooth muscle cells in the vessel wall, contributing to the development of atherosclerosis [15].

In our study, C-peptide was elevated in the blood of patients with unstable plaques, but only in obese patients, regardless of the presence of diabetes mellitus. This is probably due to the fact that according to some studies, the level of C-peptide is associated with body mass index and obesity [16,17].

TNF-α is a multifunctional proinflammatory cytokine, which is known to be involved in the pathogenesis of atherosclerosis [18]. The critical role of TNF-α in the development of atherosclerotic plaques in experimental models has been documented in detail. Numerous models of mice with TNF-α deficiency consistently demonstrated a decrease in the atherosclerotic process [19]. TNF-α can induce the production of IL-6, which stimulates the synthesis of C-reactive protein (CRP). Moreover, this cytokine can stimulate the NF-kB pathway, which regulates the expression of various genes associated with the pathogenesis of vascular diseases, such as tissue factor, VCAM-1, ICAM-1, IL-1, IL-6, and IL-8 [20]. TNF-α is also continuously expressed by adipose tissue, overexpressed in obesity, and acts as a potential mediator of insulin resistance in several animal models [21]. The main mechanism by which obesity can cause vascular changes involves perivascular adipose tissue, which can secrete TNF-α and IL-6 [22].

In our study, TNF-α was elevated in the blood of patients with unstable plaques in a subgroup of obese patients. At the same time, according to logistic regression analysis, the chance of having an unstable plaque increases with an increase in TNFa by 49% in obese patients.

IL-6 is the main pro-inflammatory cytokine produced by various cell types, including activated monocytes, macrophages, endothelial cells, adipocytes and Th2 cells [23]. IL-6 has many functions, including activation of endothelial cells, activation of the hypothalamic-pituitary-adrenal system, stimulation of lymphocyte proliferation, differentiation and oxidation of lipoproteins [24]. Due to these various effects, IL-6 may play a central role in the initiation and progression of atherosclerotic plaques [25]. Associations between IL-6 and plaque presence, plaque size, unstable plaque, including areas of reduced density, and ulceration [26], as well as carotid artery stenosis, have been documented in earlier studies. IL-6 has also been associated with the progression of carotid artery stenosis and coronary artery disease [27] in high-risk groups. In obese people, IL-6 reduces the expression of glucose transporter-4 (GLUT4) and insulin receptor substrate-1 (IRS-1) and, thus, increases insulin resistance [28].

In the study, IL-6 was elevated in the blood of patients with unstable plaques in obese patients.

Alpha cells of pancreas secreted glucagon, a peptide hormone, in response to the level of glucose and amino acids in blood plasma [7]. The results of the Naoya Osaka study suggest that glucagon may act as an anti-inflammatory agent in THP-1 cells through interaction with the glucagon receptor. Although the main mechanisms of glucagon’s anti-inflammatory effect on THP-1 cells remain unclear, glucagon has been shown to bind to the glucagon receptor and further exert various biological actions through cyclic adenosine monophosphate (cAMP) [7]. Earlier studies have shown that cAMP-boosting agents have an anti-inflammatory effect on THP-1 cells. Inflammatory effects on macrophages, such as IL-10 induction [29]. Indeed, the GLP-1 receptor agonist liraglutide, which can increase intracellular cAMP levels, imitate the anti-inflammatory effect of glucagon in THP-1 cells [30].

In our study, the level of glucagon was higher in patients with stable plaques without obesity, which could just indicate its protective properties.

Insulin directly affects the function of blood vessels. The most pronounced gas-regulating effect of insulin is stimulation of the synthesis of nitric oxide (no) and endothelin-1 (ET-1) in the endothelium, causing vasodilation and vasoconstriction, respectively [31]. Insulin stimulates the production of NO in the endothelium through the complex phosphatidylinositol-3-kinase (PI3K) pathway/protein kinase B (Act) [32]. In conditions sensitive to insulin, the vasoprotective effects of insulin are enhanced [8]. The production of NO and ET-1 balances vasodilation and vasoconstriction [31]. In insulin-resistant conditions, the PI3K pathway in the endothelium is selectively disrupted, reducing NO production [33]. Vasodilation decreases, as well as the inhibitory effect of NO on thrombosis and inflammation is weakened [34].

It is noteworthy that elevated insulin levels were more common in patients with stable plaques and without obesity, which may indicate that in conditions of insulin sensitivity, its potentially protective properties are revealed. At the same time, the chance of having an unstable plaque decreased with an increase in insulin by 3% in patients without obesity.

The limitations of the study are the small sample size, the presence of male patients only.

## 5. Conclusions

Thus, blood levels of C-peptide, IL-6, and TNFa are elevated in patients with coronary atherosclerosis in the background of obesity and with unstable plaques in the coronary arteries. In addition, in men with coronary atherosclerosis and obesity, the presence of unstable atherosclerotic plaques in the coronary arteries is directly associated with the level of TNF-α.

## Figures and Tables

**Figure 1 jpm-12-01248-f001:**
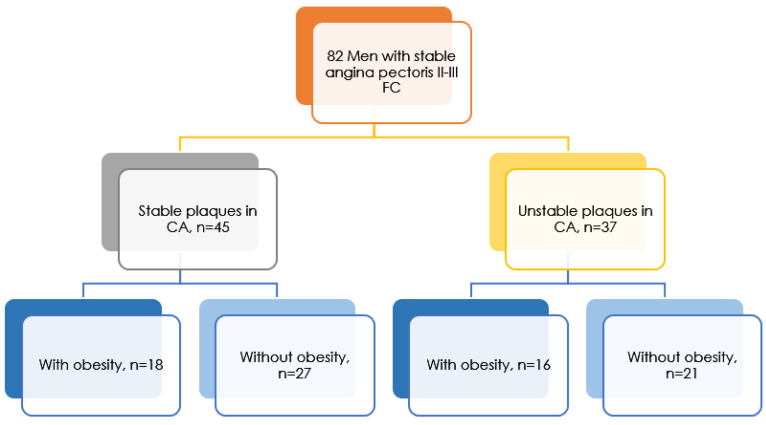
Research design.

**Figure 2 jpm-12-01248-f002:**
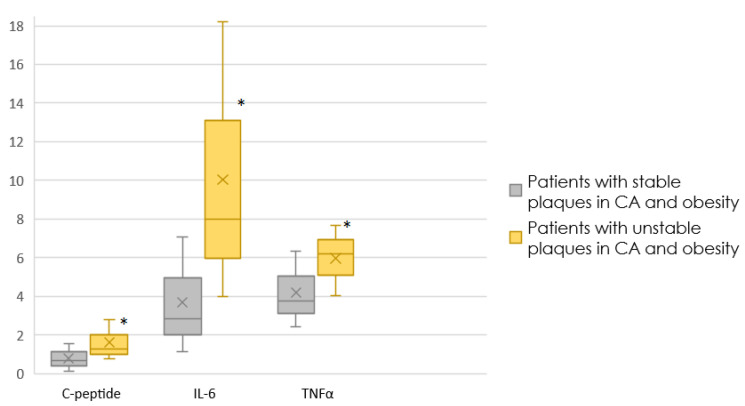
The content of C-peptide, IL-6, TNFa in the blood of obese patients, depending on the type of plaque. (*: *p* < 0.05).

**Figure 3 jpm-12-01248-f003:**
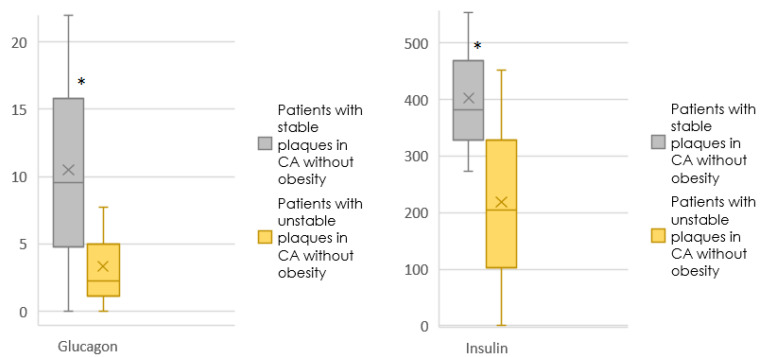
The content of glucagon, insulin in the blood of patients without obesity, depending on the type of plaque. (*: *p* < 0.05).

**Table 1 jpm-12-01248-t001:** Characteristics of population and patient groups depending on the type of plaque (stable/unstable) and the presence of obesity.

Parameter	All Patients *n* = 82	Patients with Stable Plaques in the Coronary Arteries		*p*	Patients with Unstable Plaques in the Coronary Arteries		*p*
		The Presence of Obesity *n* = 18	Absence of Obesity *n* = 27		The Presence of Obesity *n* = 16	Absence of Obesity *n* = 21	
Average age (**M ± SD**)	60.74 ± 7.16	62.20 ± 8.87	59.68 ± 5.41	0.530	59.13 ± 5.61	61.86 ± 8.20	0.182
BMI, kg/m^2^ (**M ± SD**)	29.18 ± 4.18	30.78 ± 3.85	27.27 ± 4.33	**<0.001**	32.99 ± 2.49	27.01 ± 2.45	**<0.001**
WC, cm (**M ± SD**)	92.83 ± 9.90	101.00 ± 8.16	85.05 ± 5.17	**0.031**	96.07 ± 10.00	91.58 ± 8.81	0.079
SBP, mmHg (**M ± SD**)	134.52 ± 13.36	130.90 ± 14.14	135.36 ± 14.65	0.972	137.88 ± 14.06	134.43 ± 10.19	0.397
DBP, mmHg (**M ± SD**)	82.09 ± 8.82	81.3 ± 8.44	81.68 ± 9.51	0.456	84.00 ± 9.00	81.86 ± 8.63	0.634
Smoking status (**absolute in %**)	62 (75.7%)	72.2%	66.7%	0.753	75.0%	95.2%	0.144
DM 2 type (**absolute in %**)	17 (20.7%)	11.1%	18.5%	0.684	37.5%	19.0%	0.274

**Table 2 jpm-12-01248-t002:** The content of adipokines and markers of inflammation depending on obesity and the type of atherosclerotic plaque, Me [25–75%]; pg/mL.

Parameter	Patients with Stable Plaques with Obesity *n* = 18	Patients with Stable Plaques without Obesity *n* = 27	*p*	Patients with Unstable Plaques with Obesity *n* = 16	Patients with Unstable Plaques without Obesity *n* = 21	*p*
C-peptide	0.69 [0.14; 1.54]	0.74 [0.36; 1.43]	0.379	1.26 [0.77; 2.27]	0.67 [0.33; 1.73]	**0.055**
Glucose-dependent insulinotropic polypeptide (GIP)	34.01 [22.20; 63.85]	27.51 [13.15; 49.51]	0.276	38.30 [31.85; 54.56]	16.92 [11.58; 39.80]	**0.023**
Glucagon-like peptide-1 (GLP-1)	679.79 [305.23; 880.71]	411.60 [215.67; 654.78]	0.071	693.56 [420.36; 1430.44]	444.74 [334.53; 681.92]	**0.044**
Glucagon	7.98 [0.00; 24.16]	9.58 [0.001; 22.01]	0.925	5.55 [0.00; 15.06]	2.30 [0.001; 7.77]	0.267
Interleukin-6 (IL-6)	2.86 [1.15; 7.05]	7.01 [2.81; 13.80]	0.095	7.98 [3.97; 18.20]	8.81 [1.27; 9.92]	**0.055**
Insulin	301.90 [272.95; 440.77]	382.50 [272.95; 554.28]	0.188	451.76 [272.95; 653.38]	205.02 [0.001; 451.76]	**0.006**
Leptin	6183.78 [4080.06; 12,751.17]	5416.49 [1867.87; 8509.36]	0.417	7491.86 [2167.75; 13,478.77]	3062.16 [829.62; 6196.61]	**0.021**
Monocytic Chemoattractant Protein-1 (MCP-1)	198.11 [128.49; 307.16]	185.39 [120.45; 289.62]	0.746	208.72 [160.27; 286.82]	214.99 [155.57; 249.44]	0.639
Tumor necrosis factor α (TNFa)	3.78 [2.43; 6.33]	5.52 [2.69; 7.19]	0.211	6.21 [4.02; 7.67]	4.46 [2.83; 5.87]	0.073

**Table 3 jpm-12-01248-t003:** Logistic regression analysis of the chance of an unstable plaque depending on the studied parameters.

Parameter	Model 1	Model 2 (Age-Adjusted)
C-peptide, by 1 pg/mL	1.394 (0.808–2.403), *p* = 0.232	0.994 (0.899–1.099), *p* = 0.906
TNFa, by 1 pg/mL	1.490 (1.034–2.149), ***p* = 0.033**	1.514 (1.041–2.201), ***p* = 0.030**
IL-6, by 1 pg/mL	0.996 (0.965–1.027), *p* = 0.784	0.996 (0.965–1.027), *p* = 0.795
Glucagon, for 1 pg/mL	0.951 (0.896–1.009), *p* = 0.094	0.951 (0.897–1.008), *p* = 0.092
Insulin, by 1 pg/mL	0.997 (0.994–1.000), ***p* = 0.029**	0.997 (0.994–1.000), ***p* = 0.026**

## Data Availability

Not applicable.

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
