# Peer review of "Adipokine Levels in Men with Coronary Atherosclerosis on the Background of Abdominal Obesity"

_jpm, 2022, doi:10.3390/jpm12081248_

Round 1
Reviewer 1 Report
Striukova and colleagues investigate the role of adipokines in atherosclerosis in the context of abdominal obesity. Overall, the manuscript investigate a topic of interest, even if is not a novelty. Moreover there are flaws and missing parts in the manuscript.
1. Why only male were included in the study? The authors should justify the choice.
2. Were blood samples collected fasting?
3. A brief description of the Human Metabolic Hormone V3 panel should be provided. How many metabolites? Were other adipokines available?
4. Description of blood pressure measurement is missing.
5. A table with characteristics of all the population is missing.
6. The author claim to study the abdominal adiposity, however population is not divided into categories based on their Waist. Why?
7. Dyslipidemia is mentioned, but I cannot find any analysis or descriptive statistics showing it.
8. Despite the stable/unstable plaques, are adipokines significantly different among obese or non-obese?
Author Response
Dear reviewer,
We extremely appreciate your remarks.
- The design of this study at the beginning suggested of recruiting only men because they more often are admitted to the hospital for CABG (men:women are about 3:1). This data is consisted with other studies after CABG.
After a while we started recruiting women as well, but the man:woman ratio is still unfair, so the results would have been inconclusive. So we decided to limit our study with men until we would have enough woman to compare them.
- Blood samples were collected after night fasting; we have added this information to the text.
- MILLIPLEX Human Metabolic Hormone Panel V3 is a 14-plex kit to be used for the simultaneous quantification of any or all of the following analytes in human serum, plasma, tissue/cell lysates, and culture supernatant samples: Amylin Active or Total, C-Peptide, Ghrelin Active, GIP, GLP-1 Active or Total, Glucagon, IL-6, Insulin, Leptin, MCP-1/CCL2, PP, PYY, Secretin and TNFα. We have chosen part of these parameters and included them into the study.
- Description of blood pressure measurement was added to the text.
- We extended Table 1 Characteristics of population and patient groups depending on the type of plaque (stable/unstable) and the presence of obesity with the characteristics of all the population.
- We agree that abdominal obesity as a term could be only used in cases of WC ≥ 94 cm in men (according to the 2021 ESC Guidelines on cardiovascular disease prevention in clinical practice). In our study among all patients with obesity 23 (67.6%) had also criteria of abdominal obesity – waist circumference ≥ 94 cm. So the majority of men had also abdominal obesity.
And we couldn’t consider the other 11 patients as 'metabolically healthy' patients even without this threshold of 94 cm because all of them had underwent coronary bypass surgery which made them patients of very high cardiovascular risk
Therefore, we decided to generalize this group as male abdominal obesity, since the abdominal fat of this men may be one of the organs contributing to the development of a more severe course of atherosclerosis. We have added this information to the text.
- Taking into account the very high cardiovascular risk, all patients had dyslipidemia and received statins in the most tolerable dosages. The study of dyslipidemia was not the purpose of this study and no links were obtained with biomarkers in the blood (against the background of therapy), as well as there were no differences in lipid levels between the studied groups, so we considered the presentation of these data superfluous and described it in the text in order to avoid overloading the article. We have corrected the text to avoid misunderstanding.
- Yes, we have studied groups with and without obesity, the results are below. But since the aim of our study was to examine patients with different types of atherosclerotic plaques, we consider this data redundant.
|
|
C-PEPTID |
GIP |
GLP1 |
GLUCAGON |
IL6 |
INSULIN |
LEPTIN |
MCP1 |
TNFa |
|
|
p |
,049 |
,024 |
,000 |
,103 |
,000 |
,003 |
,000 |
,126 |
,144 |
|
|
a. Obesity_gr |
||||||||||

Reviewer 2 Report
The title and content of the study is not reflected in abdominal obesity but in obesity in general. It is known that abdominal fat is the endocrine organ that mainly produces inflammatory cytokines, origin of the atherosclerotic process. The phenotype of the studied population is obese with a waist-hip measurement in centimeters and not the waist-hip ratio that according to the WHO in men should be between 0.78-0.94 . On the other hand, all the results, except BMI, are marginal (p>0.001), which could be explained by the small sample. The study is well documented but these observations must be explained to validate the title
Author Response
Dear reviewer,
We extremely appreciate your remarks.
We agree that abdominal obesity as a term could be only used in cases of WC ≥ 94 cm in men (according to the 2021 ESC Guidelines on cardiovascular disease prevention in clinical practice). In our study among all patients with obesity 23 (67.6%) had also criteria of abdominal obesity – waist circumference ≥ 94 cm. So the majority of men had also abdominal obesity.
And we couldn’t consider the other 11 patients as 'metabolically healthy' patients even without this threshold of 94 cm because all of them had underwent coronary bypass surgery which made them patients of very high cardiovascular risk
Therefore, we decided to generalize this group as male abdominal obesity, since the abdominal fat of this men may be one of the organs contributing to the development of a more severe course of atherosclerosis.
We have added this information to the text. But if the title still seems inappropriate it undoubtedly can be corrected.
